# Online Urban Waterlogging Monitoring Based on Recurrent Neural Network for Classification of Microblogging Text

Hui Liu[1], Ya Hao [1], Wenhao Zhang [1], Hanyue Zhang [1], Fei Gao [1], Jinping Tong [1]

[1]Business School, Changzhou University, Changzhou, 213159, China

5 *Correspondence to*: Jinping Tong (tjp@cczu.edu.cn)

**Abstract.** With the global climate change and rapid urbanization, urban flood disaster spreads and becomes increasingly serious in China. The urban rainstorm and waterlogging have become an urgent challenge that needs to be real-time monitored and further predicted for the improvement of urbanization construction. In this paper, we trained a recurrent neural network (RNN) model to classify microblogging posts related to urban waterlogging, and establish an online monitoring system of urban waterlogging caused by flood disaster. We manually curated more than 4,400 waterlogging posts to train the RNN model so that it can precisely identify waterlogging-related posts of Sina Weibo to timely find out urban waterlogging. The RNN model has been thoroughly evaluated, and our experimental results showed that it achieved higher accuracy than traditional machine learning methods, such as SVM and GBDT. Furthermore, we build a nationwide map of urban waterlogging based on recent two-year microblogging data.

## 1 Introduction

Due to climate change and rapid urbanization, global flood disasters have become increasingly frequent and serious, leading to traffic jams, environmental pollution, residents travel and health risks, etc.(Wheater et al., 2009; Kuklicke et al., 2016; Sofia et al., 2017). Therefore, it is crucial to address the problem of early warning and monitoring of flood disaster for the sake of life and property safety. However, it is difficult to predict nature disasters and emergencies, such as earthquake and flood, as we usually have no enough data to train an effective prediction model. Take flood as example, we expect that the model can integrate various data resources, including weather forecast, historical waterlogging events and hydrogeological data, to predict the occurrence of flood and timely remind the residents to prepare for the disaster, thereby reduce the loss and achieve the early warning function.

Existing methods for flood disaster early warning generally use meteorological and hydrological data as the basis of construction of forecasting models. Researchers build hydrological models that take into account various factors, and simulate the occurrence, progression and consequence of flood disaster (Tawatchai et al., 2005; Yu et al., 2015; Anselmo et al., 1996; Lima et al., 2015). Subsequently, multilevel thresholds corresponding to different warning levels are determined based on the simulation result of the hydrological models. Also, some scholars have developed methods for early warning of flood disaster. For example, Xiao et al. developed a flood forecasting and early warning method based on similarity theory

and hydrological model to extend the lead time and achieve dynamic rolling forecasting (Xiao et al., 2019). Kang et al. proposed a flood warning method based on dynamic critical precipitation (Kang et al., 2019). In order to meet the actual need of flood early warning, Liang et al. constructed a new "grade-reliability" comprehensive evaluation of accuracy of flood early warning. The method combines the grade prediction accuracy evaluation criteria and uncertainly analysis method to evaluate the reliability of the predicted results (Liang et al., 2019). Also, Liang discussed how to determine the flood early warning and forecasting time by use of rising rate analysis on the basis of the flood rising rate changing over time in historical data (Liang, 2019).

The rapid development of mobile internet and smartphone has boosted various social media, such as Weibo and Twitter. Weibo is run by sina.com Company in China and it is similar to Twitter. A Weibo post contains the fields such as user ID, user name, microblog content, posted time, etc. Some posts also include pictures and video. In fact, Weibo has become very popular in Chinese people, and accordingly it becomes an important information source of flood and nature disaster (Robinson et al., 2014). Some researchers explored social media to extract information about disasters. For example, de Bruijn et al. proposed a database for detecting floods in real-time on a global scale using Twitter. This database was developed using 88 million tweets, from which they derived over 10,000 flood events (de Bruijn et al., 2019). Cheng et al. used Sina-Weibo data to reveal the public sentiments to nature disasters on social media in the context of East Asian culture. The Pearson correlation between information dissemination and precipitation was analyzed, and important accounts and their information in social networks were determined through visual analysis (Cheng et al., 2019). Barker et al. developed a prototype of national-scale Twitter data mining pipeline for improved stakeholder situational awareness during flooding events across Great Britain, by retrieving relevant social geodata, grounded in environment data sources (flood warnings and river levels) (Barker et al., 2019). Wang et al. analyzed the subject words and user sentiments of the earthquake events based on the week-long discussion on Sina Weibo after the earthquake in Japan (Wang et al., 2012). Zhang et al. used the Shanghai Bund Stampede incident as an example, according to the response time, response speed, microblog contents and microblog interaction of government microblog after the emergency, to analyze and evaluate the information release and response ability of government in emergency. Some concrete ways and suggestions for the government to make information release more effective were put forward (Zhang et al., 2015).

Text classification is a hot topic in the field of Nature Language Processing (NLP) (Hu et al., 2018; Kim, 2014), and has been widely used to identify important information of interest from social media. Amin et al. proposed a novel model for Dengue disease detection based on social media alone. Their RNN model adopts word embedding techniques, including TF-IDF and n-gram, to capture broad context of the words in social media text for better classification (Amin et al., 2020). Jelodar et al. used automated extraction of the novel coronavirus-related posts from social media and topic modeling to uncover various issues related to COVID-19 from public opinions, and then investigated how to use LSTM recurrent neural network for sentiment classification of COVID-19 comments (Jelodar et al., 2020). In this paper, we employed text classification of Weibo posts to identify urban waterlogging caused by flood. By manually collecting more than 4,400 waterlogging-related Weibo posts from 2017 to 2018, we built a gold-standard dataset to train a text classification model.

We tested three popular models, including recurrent neural network (RNN), support vector machine (SVM) and gradient boosting decision tree (GBDT), and found that RNN achieved best performance, evaluated on an independent test set that contains 400 Weibo posts (positive and negative) of 2019. Furthermore, we built a nationwide map of urban waterlogging based on recent two-year microblogging data, and a monitoring system based on WeChat applet that will alert the user via voice alarm when he/she approaches a waterlogging point.

As far as our knowledge, this is the first manually validated and the largest dataset of Weibo posts related to urban flood deposits, which is ready for building text classification model to identify Weibo posts that truly reports waterlogging events. Also, we are the first to build a nationwide map with more than 6,000 waterlogging points, which covers most cities in China. Furthermore, the RNN model trained on our dataset can precisely identify flood deposits via online Weibo classification, and our monitoring system based on WeChat applet would effectively benefit users to reduce the risk and loss caused by flood.

## 2 Materials and methods

The overall framework of our method is shown in Figure 1. There are four steps to build the RNN-based text classification model of Weibo posts for monitoring urban flood waterlogging. First, we collect 2017-2018 Weibo posts, and get more than 70 thousand posts after deduplication. After Chinese word segmentation and removal of stop words, we build feature representations of each Weibo posts based on word vector space and word2vector model. Next, three popular text classification models are trained and tuned for their respective parameters, and their performances are evaluated on an independent test dataset that is manually collected from 2019 Weibo posts.

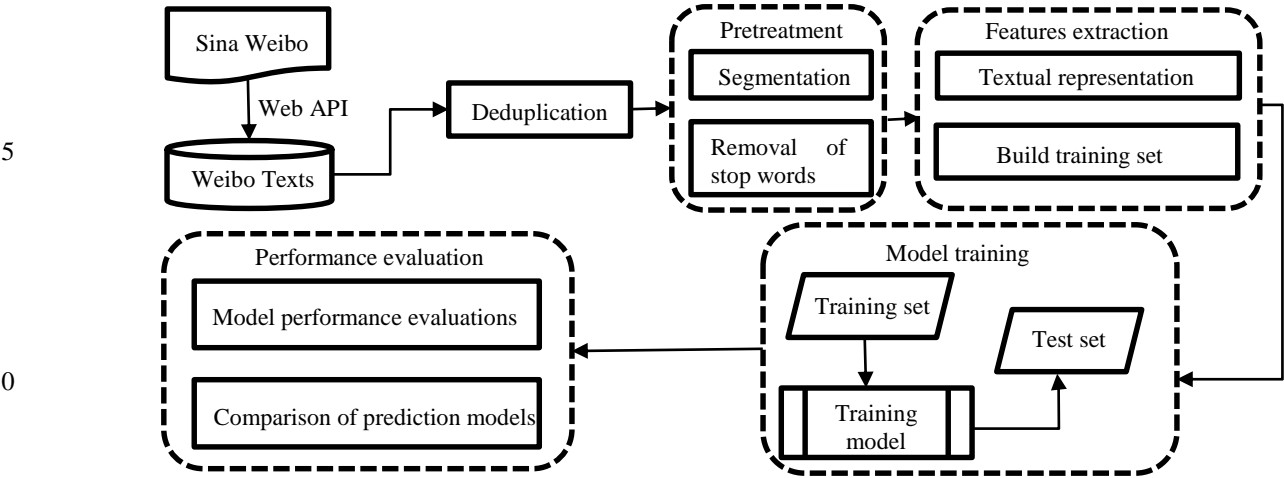

**Figure 1:** Illustrative diagram of our method framework

## 2.1 Data sets

### 2.1.1 Data sources

The Weibo posts were obtained using Sina Weibo API. To exclude completely irrelevant posts, we downloaded only the Weibo posts including keywords "淹" or "积水" (drowning or waterlogging in Chinese). All microblogging text were saved in a comma-separated values (CSV) file to form data table, including the issuer, microblogging text, post time, etc. We found that there were some posts with attached coordinates. However, not every post had attached coordinates, because the GPS function is often turned off by users (de Bruijn et al., 2018).

For the sake of geographical localization of urban waterlogging points from the content of Weibo posts, we also downloaded the catalog of nationwide communities of 307 cities in China, including community name, geographical location, floor area ratio, greening rate and other information, from a famous housing website anjuke.com in China. Each city includes an average of 1,500 communities.

### 2.1.2 Data cleaning

Weibo posts have a large number of repetitions, due to its commenting function with the forward of original text and image content. A hot post may have been commented for many times, but the main body of these Weibo posts is same. Another case is that "@ (retweet)" function also leads to many duplicates. We removed duplicates by using a string match pattern that compared a number of leading characters of two posts. For example, if the first 15 characters of two posts were same, they were considered as duplicates and kept only one.

We suppose downloaded posts stated the fact that waterlogging events occurred in some places. However, in real life, more complex situations do occur, such as some places with heavy rainfall but not flooding, indicating that local drainage infrastructure and dewatering facilities works well and the residents were not threatened by flood. On the other hand, a large part of Weibo posts were actually irrelevant to flood and waterlogging. For example, many posts contained the keywords of Chinese drowning or waterlogging mentioned above, but discussed about some diseases such as "hydrocephalus", "knee

dropsy" and so on. Such Weibo posts had nothing to do with our task. To exclude these specious posts, we manually marked each post positive or negative label based on whether the content of the posts were related to urban waterlogging.

### 2.1.3 Localization of flood deposits

First, the Weibo posts that contained words like "certain community" and "highland" were excluded, as the location of the flood deposits mentioned in these posts were difficult to be determined.

Second, most posts have no attached coordinates and some posts related to urban waterlogging actually specified the locations of the flood deposits in the text, instead of the coordinates attached to the posts. For example, the post "Xingqing district, Yinchuan city, Shanghai road (Jinning Street to Fenghuang Street) section because of the heavy rain, the road surface water was more serious." specified the waterlogging occurred in Shanghai road in the text of this post. We extracted

the terms about communities, roads and orientation from the posts. Subsequently, we used the catalog of communities of 307 cities in China to match these terms so that we could determine the geographic location of the flood deposits reported via these Weibo posts. As a result, we cannot determine the location of the flood deposit using coordinates attached to the post. The localizations of the waterlogging were done based on the textual content. For this purpose, we have no deal with two problems: one was that a community name contained the microblog overlapped with other communities, namely, two or more communities had the same name but located in different cities. The other was that the communities mentioned were not included in the catalog. For both cases, we exerted to manual matching and deduplication. We manually checked the communities with duplicate names in different cities to ensure that the flood deposits were accurately located. Also, if a post referred to certain city not included in the catalog, we manually identify the waterlogging locations by exploiting the community names or the coordinates attached to the post.

Finally, we deleted duplicate posts that had same text and same location. If their content was different, we would keep all, even if their location was same. For example, we have two posts: (i) On the morning of May 12th, the reporter learned from the Nanning Traffic Police Detachment that because of the rainy day leading to Pingle Avenue surrounding roads had part of water. (ii) At present, there have been water accumulation in some sections of Yinhai Avenue, Ruihejiayuan and Pingle Avenue, please pedestrians and vehicles are requested to pass carefully. We kept both posts. If the content of two or more posts was the same, we keep only one post. In total, we got 4,451 Weibo posts that were successfully located to urban flood deposits. Some examples are shown in Table 1.

**Table 1:** Examples of Weibo posts related to urban flood deposits

| 微博内容 | 时间 | 省份 | 市 | 地点 |
|---|---|---|---|---|
| #社区工作#昨天大雨造成工农 | 2017 年 5 月 5 日 | 江苏 | 南京 | 工农新村 317 号 |
| #南充身边事#【积水困住公交 | 2017 年 5 月 5 日 | 四川 | 南允 | 高坪区高都路 |
| #九江身边事#【九江八里湖新 | 2017 年 5 月 5 日 | 江西 | 九江 | 九江八里湖新区通湖路铁路 |
| #直播莆田#【市区路面积水网 | 2017 年 5 月 6 日 | 福建 | 莆田 | 涵江沃尔玛门口红路灯处 |
| 【福州金山九条"金"字路多 | 2017 年 5 月 6 日 | 福建 | 福州 | 金榕北路马榕小区 |
| #今日早报#【一场大雨，南宁 | 2017 年 5 月 7 日 | 南宁 | 南宁 | 五一路铁路桥底 |

**Table 1\*:** Examples of Weibo posts related to urban flood deposits

| Weibo content | Time | Province | City | Location |
|---|---|---|---|---|
| #Community Work# | May 5, 2017 | Jiangsu | Nanjing | No. 317, Gongnong new village |
| #Nanyun side things# | May 5, 2017 | Sichuan | Nanyun | Gaodu road, Gaoping district |
| #Jiujiang side things# | May 5, 2017 | Jiangxi | Jiujiang | Jiujiang bali lake new district Tonghu road |
| #Live broadcast of | May 6, 2017 | Fujian | Putian | Hanjiang Wal-mart at the gate of the red street |
| [The fuzhou jinshan | May 6, 2017 | Fujian | Fuzhou | Jinrong north road, Marong community |
| #Today morning# [a | May 7, 2017 | Nanning | Nanning | Under Wuyi railway bridge |

* Table 1* is a translation of Table 1.

### 2.1.4 Selection of negative samples

The positive samples were exactly those posts that included both locations and waterlogging. The negative samples consisted of all remaining posts, including those irrelevant to waterlogging keywords, as well as the posts with waterlogging keywords but without specific locations in the text. For example, the posts that actually discussed diseases, such as hydrocephalus, keen dropsy and so on, were regarded as negative samples.

    In addition, it was worth noting that a huge number of posts irrelevant to waterlogging actually included a lot of

location names, and these posts were used as negative samples to train the classifier. So, we believed that the classifier could actually learned the waterlogging-related features of posts. During the selection of negative samples, the positive samples were excluded. Finally, we built a training dataset that includes 4,451 positive samples (labeled as 1) and 246,341 negative samples (labeled as 0).

### 2.2 Extract feature vectors and construct training set

To train a text classifier, it is necessary to transform a Weibo post, which typically is strings of words, into a feature vector suitable for classification tasks. The first step of preprocessing Weibo posts included Chinese word segmentation and removal of stop words. Thereafter, we constructed a bag of words by extracting unique words in the training set, and then built feature representations of each Weibo post based on word vector space and word2vector model.

### 2.2.1 Data preprocessing

Due to the Sina Weibo posts are written in Chinese, the text does not have a natural separator between words. Therefore, it is necessary to perform Chinese word segmentation on Weibo posts, which is actually a basic process in Chinese natural language processing. We used the Jieba tool to segment words of Weibo posts.

    There were many common auxiliary words, prepositions, and so on in Chinese, such as "的 (of)" and "在 (in)", which should be got rid of with the help of dictionary of stop words. In the word-bases retrieval system, words with high frequency

but without retrieval significance are determined as stop words. We completed this task by exploiting the stop word list released by Harbin Institute of Technology (Guan et al., 2017), which is a widely used stop word catalogue. Also, many words were useless to our task but appeared in the Weibo posts, such as "视频 (video)", "微博 (Weibo)", etc. These words were also added to the stop words dictionary so as to remove such words from the Weibo posts. For example, for the sentence "昆山昨天大雨，新城域被淹，房价还比周边没被淹的小区贵，真是刺裸裸讽刺！("Kunshan heavy rained

yesterday, the new city domain was flooded, and the house price was more expensive than the surrounding flooded communities. It is really naked irony!") ". After Chinese word segmentation and removal of stop words, the result was as follows: "昆山 昨天 大雨 新城 域 被淹 房价 比 周边 被淹 小区 贵 刺裸裸 讽刺 (Kunshan | Yesterday | Heavy rain | New city | Domain | Flooded | House price | Than | Surrounding | Flooded | Community | Expensive | Naked | Irony)".

### 2.2.2 Feature representation

The purpose of feature representation is to encode a numeric vector that represents the content of a Weibo post. We considered two most popular methods, TF-IDF and word2vector. TF-IDF (Term Frequency-Inverse Document Frequency) is the most popular term-weighting scheme for information retrieval and data mining. The word2vector models the context of words and the semantic relationship between words and context, and maps words to a low-dimensional real number space to generate the corresponding word vectors (Wang et al., 2018). This paper used both TF-IDF and word2vector methods for

feature representations of Weibo posts. TF-IDF was used to build feature vectors prepared for input into SVM and GBDT classifiers, while word2vector was used for RNN, SVM, and GBDT classifiers.

TF-IDF scheme is a statistical method employed to evaluate the importance of a word in a document. TF is the word frequency, formally written as $tf(t,d)$, which means the frequency of the term $t$ appears in the document $d$, and reflects the correlation between $t$ and $d$. IDF is the inverse document frequency, formally written as $idf(t)$, which represents the

quantification of the term distributions in a collection of documents. The commonly used calculation method is $\log(\frac{N}{n_t} + 0.01)$, in which $N$ represents the total number of documents, and $n_t$ represents the number of documents in which term $t$ appears (Xiong et al., 2008). TF-IDF represents the importance of relevant terms in the document space. The calculation method as follows Eq. (1) (Salton et al., 1998):

$$w(t,d) = tf(t,d) * log(\frac{N}{n_t} + 0.01), \qquad (1)$$

where $w(t,d)$ represents the weight of term $t$ in document $d$.

The larger the TF-IDF of a term, the higher the importance in the document. By calculating the TF-IDF value of each word in a Weibo post, we can construct a real-value vector representation of this post. This paper used the TfidfVectorizer in scikit-learn Python package to calculate the TF-IDF value of each word of Weibo posts. According to the TF-IDF value of words, the first 5,000 words were selected to construct the dictionary and subsequently each post was converted into a 5,000-

dimensional vector.

Word2vec exploits the idea of deep neural network to simplify the processing of text content into vector representation in $K$-dimensional space, and the similarity in vector space can be used to represent the semantic association between words (Mikolov et al., 2013). Word2vec takes as its input a large corpus of text to produces a vector space, and assigns each unique word a distributed representation in the space.

We used the Gensim library (Gesim, 2019), which takes as input the urban waterlogging-related Weibo posts, to train a 100-dimensional word2vector model with the skip-gram to obtain the vector representation of each word. The structure of the skip-gram model is shown in Figure 2. Its underlying rationale is that given a certain word to predict the context. The specific calculation method is Eq. (2):

$$W_t = W_{t-c},\cdots,W_{t-1},W_{t+1},\cdots,W_{t+c}, \qquad (2)$$

where $W_t$ is the current word and $c$ is the size of the window. After learning with a large corpus of Weibo posts, the weights

were the corresponding vector representation for each word to train RNN model (Feng et al., 2018).

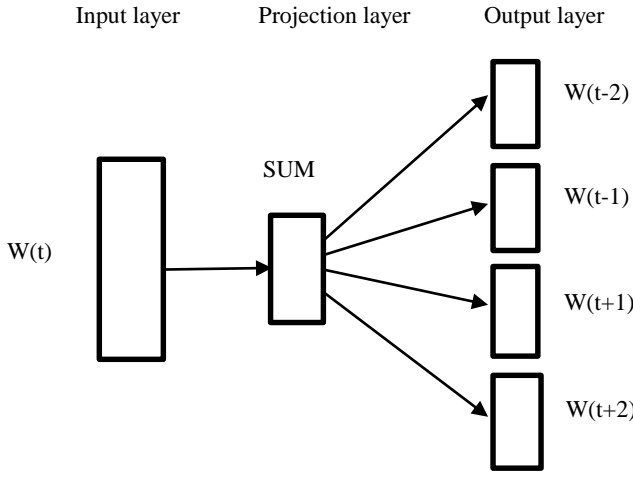

**Figure 2:** Skip-gram Model

### 2.2.3 Undersampling

Our whole dataset contains 246,341 negative samples and 4,451 positive samples. As the number of negative samples
intensively surpasses that of positive samples in the training dataset, the imbalance often leads to ill-structured decision
boundary of classification that is overwhelmed by the majority class and ignores the minority class (Chawla et al., 2004). For
example, in a case of an imbalance level of 99, a classifier that minimizes the error rate would decide to classify all examples
into the majority class to achieve a low error rate of 1%, but in fact all minority examples are misclassified. Therefore, the
imbalance problem must be handled carefully to build a robust classifier in a problem with a large degree of imbalance (Liu
et al., 2008).
We adopted the undersampling technique to reduce the number of examples belonging to the majority class with the
objective of equalizing the number of examples of each class (Garcá et al., 2009). The undersampling is done by using a
third-party python package imblearn. The RandomUnderSampler function in the imblearn package implements boostrap
sampling by setting the parameter replacement to true. This function randomly removes samples in original dataset with
multiple classes to build a balanced subset of the whole samples. In particular, we considered the negative samples as the
majority class and the positive samples as the minority class. We used the undersampling method to resample same number
of negative samples as positive samples, and then combined them to create a balanced dataset to train the classification
model. The undersampling process was repeated for enough times to ensure that every sample would be seen by the
classifier.

## 2.3 Training of classifiers

We tested three popular models, including recurrent neural network (RNN), support vector machine (SVM) and gradient boosting decision tree (GBDT), using both TF-IDF features and distributed representation derived from word2vector algorithm.

### 2.3.1 Recurrent neural network (RNN)

The most commonly used in natural language processing is the recurrent neural network. Recurrent neural network (RNN) is
used to process sequential data, which takes sequence data as input, and performs recursion in the evolution direction of sequence, and all nodes (recurrent units) are connected by chain. The recurrent neural network and its unfolding structure are shown in Figure 3 (LeCun et al., 2015). The RNN introduces a directional loop to pass down the parameters of the hidden layers state $S_{t-1}$ at the previous moment and calculate the hidden state $S_t$ at the next moment, so as to achieve the persistence of information and solve the problem of association between the input information before and after (Liu et al.,
2018).

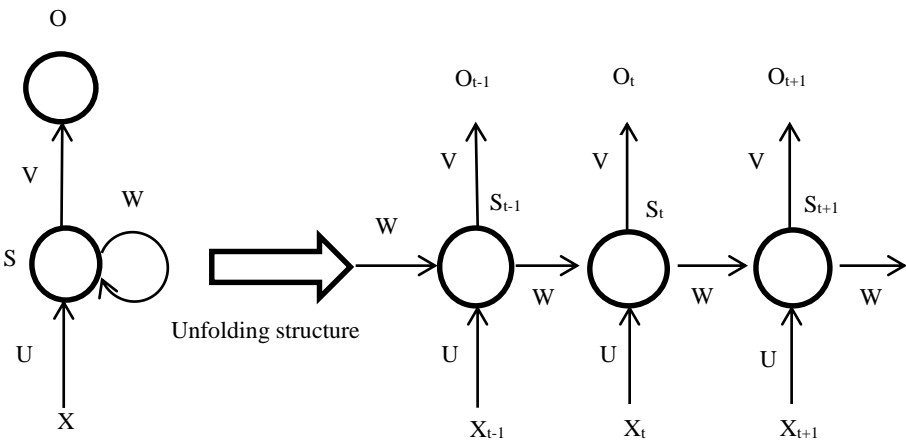

**Figure 3:** RNN and its unfolding structure

where $X$ is the input layer, $O$ is the output layer, $S$ is the hidden layer. $V$, $W$, and $U$ are the weights of the input layer, hidden
layer, and output layer respectively, and $t$ is the number of calculations for the $t$-th time. Calculating the hidden state for the $t$-th time as follows Eq. (3):

$$S_t = f(U * X_t + W * S_{t-1}), \qquad (3)$$

In this way, the current hidden layer calculation results and the current input are related to the previous hidden layer calculation results, and the purpose of the memory function is achieved.

Unfortunately, it is difficult for the RNN model to learn long-distance correlation information in a sequence, which will affect the classification effect. Therefore, this paper adopted the improved RNN model by the LSTM. Long short-term memory network (LSTM) (Hochreiter et al., 1997) is a special RNN that can learn long-distance dependent information. The key design of LSTM is the state of cells that throughout the entire network. The "gate" structure is used to remove or add information to the cell state, thereby updating the hidden state of each layer. In this paper, we used the improved RNN model

by the LSTM to replace each hidden layer with a cell that has memory function. The LSTM has great advantages in processing time series and language text sequences (Niu et al., 2018). The network structure of LSTM is shown in Figure 4.

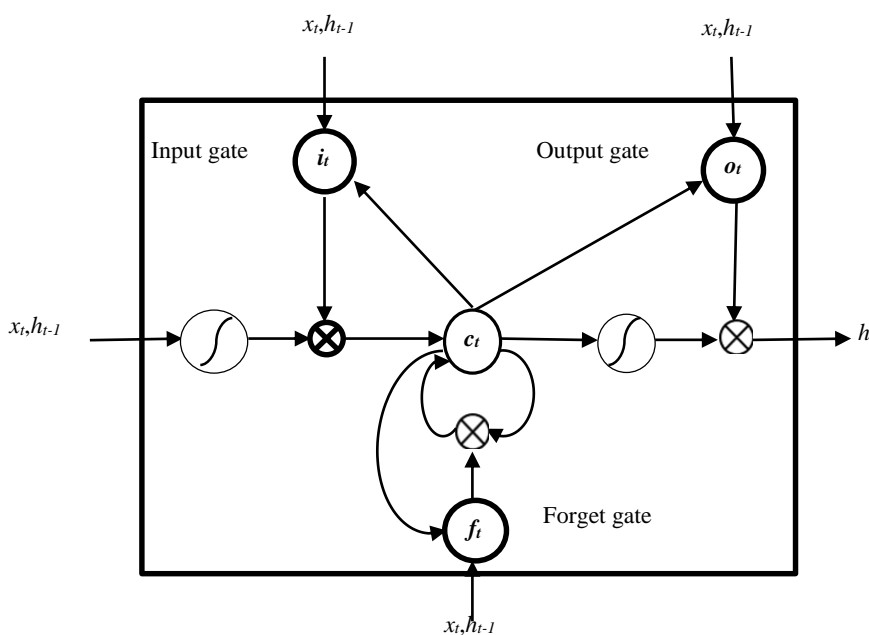

**Figure 4:** The structure of LSTM

Each unit of the LSTM model is composed of the input gate $i$, the output gate $o$, the forget gate $f$, and the internal memory unit $c$. Selectively input, passed and forgotten operation for internal memory through these three thresholds, which can effectively overcome the vanishing gradient problem (Li et al., 2018). Intuitively, the forget gate controls the amount of

which each unit of the memory cell is erased, the input gate controls how much each unit is updated, and the output gate controls the exposure of the internal memory state (Liu et al., 2016). The hidden state of the LSTM model at time $t$ as follows Eq. (4):

$$i_t = \sigma(x_t U^i + h_{t-1} W^i + b^i) \quad f_t = \sigma(x_t U^f + h_{t-1} W^f + b^f)$$

$$o_t = \sigma(x_t U^o + h_{t-1} W^o + b^o) \quad g_t = tanh(x_t U^g + h_{t-1} W^g + b^g) \qquad (4)$$

$$c_t = c_{t-1} \otimes f_t + g_t \otimes i_t \qquad h_t = tanh(c_t) \otimes o_t$$

where $\sigma(\cdot)$ denotes the sigmoid activation function, $\otimes$ denotes element-wise multiplication, $W^*$ and $U^*$ are the weight matrix in the network, $b^*$ is the bias term, $i_t$, $o_t$, and $f_t$ are the values of the input gate, output gate and forget gate at time $t$ respectively (Li et al., 2018).

During the experiments, the input layer of the RNN model imported the word vector representation matrix of the words in the sentence. For example, there are $n$ words in a microblog post, and the dimension of the word vectors is $k$, the size of the input matrix is $n*k$. Then the "gate" structure of the LSTM model removes or adds information to the cell state of the network to update the hidden state of each layer. The hidden state is input to the softmax layer, and next to output the final classification results, thereby realizing text classification.

### 2.3.2 Support vector machine

The support vector machine (SVM) classifier is widely used to solve two-category problems. The SVM model is defined as a linear classifier with the largest interval in the features space. Its basic idea is to find the classification hyperplane that can divide the training data set correctly and has the largest geometric interval (Tan et al., 2008). The SVM model is suitable for classification in high-dimensional space and has a good performance on small-size data sets.

### 2.3.3 Gradient boosting decision tree

Gradient boosting decision tree (GBDT) is a boosting algorithm proposed by Friedman (Friedman, 2000) in 2001. GBDT is an iterative decision tree algorithm, which is composed of multiple decision trees, and the answers of all trees are added up to make the final decision. The model established each time is in the gradient descent direction of the previous established model loss function, so that it performs better than traditional boosting algorithms, which reweight the correct and wrong samples after each round of training (Feng et al., 2017).

## 3 Experiment

### 3.1 Confusion matrix

The confusion matrix, also known as the possibility table or error matrix, is a specific matrix used to present the visual effect map of the classifier performance. The rows represent the predicted values, whereas the columns represent the actual values. The categories used in analysis are false positive, true positive, false negative, and true negative. The structure of the confusion matrix is shown in Table 2.

**Table 2:** The structure of confusion matrix

| Confusion Matrix | | Actual Classes | |
|---|---|---|---|
| | | Positive | Negative |
| **Predicted Classes** | Positive | TP | FP |
| | Negative | FN | TN |

For example, a post reads as follows: In the early hours of this morning, there was a heavy rain in Shenzhen. Many roads were flooded, and only the top of cars could be seen. Among them, the water at the bottom of Hezhou Bridge was serious, which caused the road to be interrupted. TP means true positive: the truth is positive, and the classifier predicts a positive. For example, the bottom of Hezhou Bridge is flooded, and the classifier accurately reports this. TN means true negative: the truth is negative, and the classifier predicts a negative. For instance, the bottom of Hezhou Bridge is not flooded, and the classifier accurately reports this. FP means false positive: the truth is negative, but the classifier predicts a positive. Such as the bottom of Hezhou Bridge is not flooded, but the classifier inaccurately reports that it is. FN means false negative: the truth is positive, but the classifier predicts a negative. For example, the bottom of Hezhou Bridge is flooded, but the classifier inaccurately reports that is not.

## 3.2 Hyperparameters optimization

When building a deep learning model, the selection of hyperparameters is very important. This paper used a grid search to find the optimal hyperparameters of the RNN. The grid search is an optimization strategy by specifying parameter values to exhaustive enumerations to select optimal parameters. The hyperparameters to be selected include optimizer, batch size, and keep probability and so on. To reduce the computational overhead, we chose to fix other parameters, and changed the parameter value for experimentation.

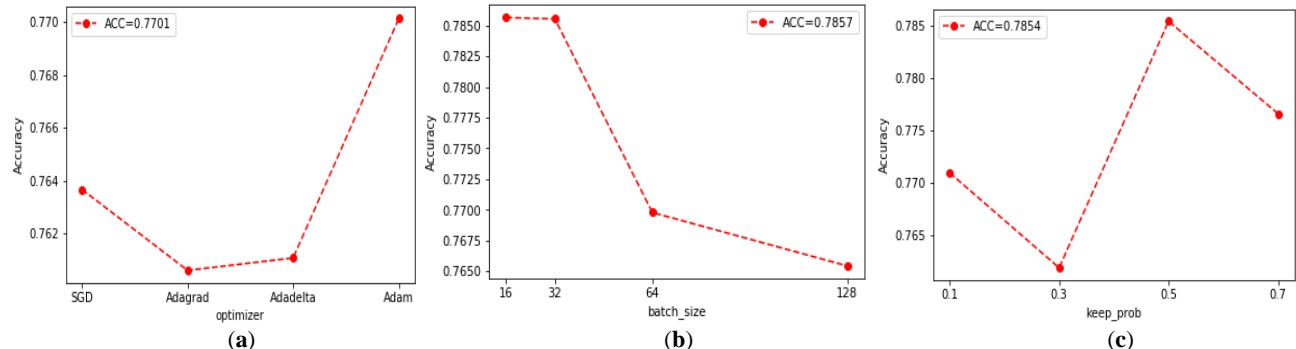

**Figure 5:** Hyperparameters selection based on Grid-Search: optimization functions selection image (**a**), batch_sizes selection image (**b**), keep_probs selection image (**c**).

Optimizer is used to update weights when training a deep neural network. We tested several adaptive Optimizers, including SGD, Adagrad, Adadelta and Adam. As can be seen from Figure 5a, the Adam optimizer achieve the highest accuracy by 77%. Therefore, we select Adam as the optimizer in the subsequent performance evaluation experiments.

Batch size represents the amount of samples captured in one training round, and its value affects the training speed and model optimization. As shown in Figure 5b, the accuracy of the RNN model reduces but still remains above 0.76 with the value of batch size ranging from 16 to 128. When the batch_size is 16, the accuracy reaches the highest 0.7857. Therefore, the size of the batch size is set 16.

We adjusted the size of keep probability to prevent overfitting and improve the generalization ability of the model. As can be seen from Figure 5c that when keep probability is equal to 0.5, the RNN model reaches the highest accuracy 0.7854. Therefore, the size of keep probability is set to 0.5.

In addition, the learning rate and epoch are tuned to 0.001 and 5, respectively. The selection of important parameters are shown in Table 3.

**Table 3:** Parameters setting of RNN model

| Parameters | Value |
|---|---|
| Optimizer | Adam |
| batch size | 16 |
| keep probability | 0.5 |
| learning rate | 0.001 |
| epoch | 5 |

## 3.3 Verification of flood deposits based on AutoNavi waterlogging map

In cooperation with the Public Meteorological Service Center of China Meteorological Administration, AutoNavi Map, a famous online map service provider in China, has released a nationwide map of urban waterlogging based on AutoNavi inherent road and traffic data, historical flood deposits reported by traffic polices. AutoNavi map App visualizes the urban flood deposits that can be retrieved by users.

We collected the flood deposits in AutoNavi waterlogging map, including the degree of floods, the latitude and longitude of each flood deposits. Next, we selected the flood deposits in the Nanjing city in AutoNavi waterlogging map to check how many the flood deposits identified from Weibo posts overlapped.

## 4 Experiment

## 4.1 Performance measures

We adopted a variety of measures to evaluate the performance of our proposed method, including accuracy (ACC), precision (P), recall (R), and F1-measure (F1). The receiver operating characteristic (ROC) curves and Area Under Curve (AUC) were

also used as criteria for performance evaluation. The accuracy (ACC) is defined as the ratio between the correctly classified samples to the total number of samples as in Eq. (5). The precision (P) represents the proportion of positive samples that are correctly classified to the total number of positive samples as in Eq. (6). The recall (R) is expressed as the ratio of the correctly classified negative samples to the total number of negative samples as in Eq. (7). F1-measure (F1), also refers to as F1-score, represents the harmonic mean of precision and recall as in Eq. (8) (Tharwat, 2018). Higher the value of F1, better the performance of the method. The ROC curve is a graphical plot that illustrates the diagnostic ability of a binary classifier system as its discrimination threshold is varied. The curve is created by plotting the true positive rate (TPR) against the false positive rate (FPR) at various threshold settings (Receiver, 2020). The AUC metric calculates the area under the ROC curve (Tharwat, 2018). The higher the AUC value, the better the performance (Yu et al., 2014).

$$ACC = (TP + TN)/(TP + FN + FP + TN), \qquad (5)$$

$$P = TP/(TP + FP), \qquad (6)$$

$$R = TP/(TP + FN), \qquad (7)$$

$$F1 = 2 * (P * R)/(P + R), \qquad (8)$$

## 4.2 Performance evaluation

### 4.2.1 Performance evaluation on validation set

We tested two types of feature representations, TF-IDF and word2vec, on three classifiers, including SVM, GBDT, and RNN. Note that the RNN model cannot be applied to TF-IDF feature representation as its input requires sequences. 200 samples were randomly selected from the positive and negative samples sets respectively as the validation set (400 samples in total), and the remaining samples were used as the training set. The performance measures were computed based on the predicted result of validation set by learned models. The process was repeated for 50 times, and the averages were reported as the final performance measures.

**Table 4:** Validation set results of SVM, GBDT, and RNN models

| Method | ACC | R | P | F1 | AUC |
|---|---|---|---|---|---|
| GBDT+ TF-IDF | 0.86 | 0.73 | 0.98 | 0.83 | 0.96 |
| GBDT+word2vec | 0.81 | 0.63 | 0.98 | 0.76 | 0.91 |
| SVM+TF-IDF | 0.88 | 0.80 | 0.96 | 0.87 | 0.96 |
| SVM+word2vec | 0.87 | 0.77 | 0.96 | 0.85 | 0.95 |
| RNN+word2vec | 0.96 | 0.99 | 0.93 | 0.96 | 0.99 |

As shown in Table 4, the RNN classifier based on word2vec feature representation achieved the best performance. The accuracy, recall, and F1 value of this model reach above 96%, and the AUC value also achieves 0.99, indicating that the RNN is an effective model for online classifying waterlogging-related posts. According to the experimental results, it is

found that the RNN model based on word2vec features performs generally better than traditional classifiers, such as GBDT+TF-IDF (Acc=0.86), GBDT+word2vec (Acc=0.81), SVM+TF-IDF (ACC=0.88) and SVM+word2vec (Acc=0.87).

## 4.2.2 Performance evaluation on Independent set

To further verify the effectiveness of the RNN model, we built an independent test set to evaluate the model. The independent test set contains 400 Weibo posts (200 positive and negative, respectively) of 2019. These posts were downloaded using the keywords "淹" or "积水" (drowning or waterlogging in Chinese) and then manually checked. Each trained model was performed on the certain independent test set and the performance measures were computed, as shown in Table 5 and Figure 6. The RNN model based on word2vec feature significantly outperforms all other models and achieves highest accuracy, recall, and F1 values. It also has achieved the largest AUC value 0.95.

**Table 5:** Test set results of SVM, GBDT, and RNN models

| Method | ACC | R | P | F1 | AUC |
|---|---|---|---|---|---|
| GBDT+ TF-IDF | 0.77 | 0.62 | 0.89 | 0.73 | 0.83 |
| GBDT+word2vec | 0.83 | 0.69 | 0.97 | 0.81 | 0.93 |
| SVM+TF-IDF | 0.81 | 0.72 | 0.88 | 0.79 | 0.90 |
| SVM+word2vec | 0.86 | 0.88 | 0.84 | 0.86 | 0.95 |
| RNN+word2vec | 0.90 | 0.97 | 0.85 | 0.91 | 0.95 |

Furthermore, it can also be seen that the word2vec feature representation effectively improves the performance of text classification models on the independent test data set. This shows that the word vectors generated by word2vec model are more informative than traditional TF-IDF in expressing the features of microblogging posts related to urban waterlogging. For both GBDT and SVM, the models trained on word2vec features got better performance on the independent test dataset, compared to those trained on TF-IDF. In fact, we find similar results in Table 4, RNN+word2vec obtained the best performance, followed by SVM+word2vec.

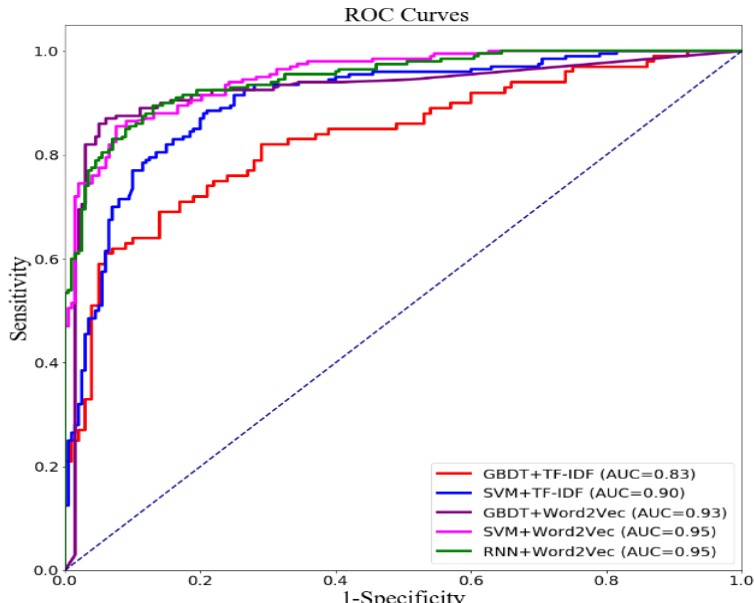

**Figure 6:** ROC curves of three models

**4.3 Verification based on AutoNavi waterlogging map**

We collected 1,660 flood deposits from AutoNavi waterlogging map, and found that 75 flood deposits locate in Nanjing city. Among the 165 flood deposits in Nanjing derived from Weibo posts of 2017-2018, there were 18 overlapped points on the AutoNavi waterlogging map. Figure 7 shows these overlapped flood deposits.

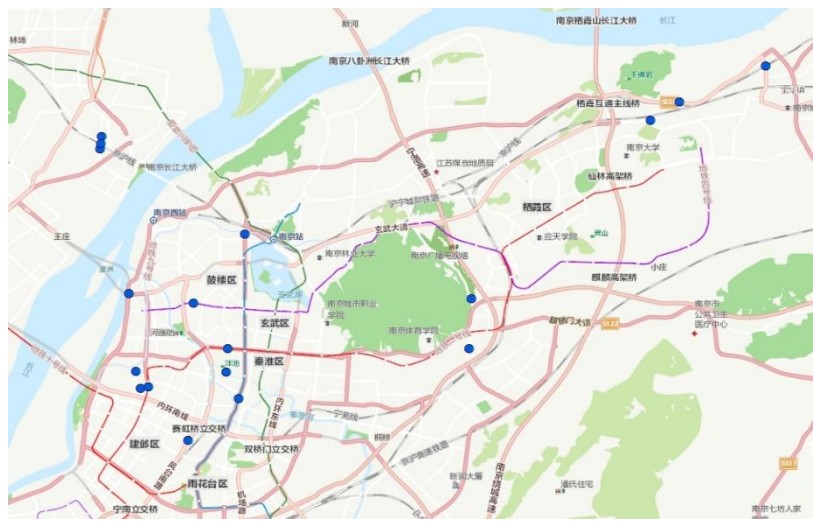

**Figure 7:** Overlapping flood deposits in Nanjing (© Google Maps 2019)

To verify the effectiveness of our RNN model, we used the locations of these 18 overlapped flood deposits as keywords, such as "毕升路 (Bisheng Road)", to retrieve posts from Sina Weibo and got more than 800 microblogging posts, among

which only 29 posts were really related to these flood deposits. Taking all these posts as input, the trained RNN model achieved 0.836 accuracy. This experiment confirmed that the RNN model proposed in this paper can applicable for online monitoring of urban waterlogging. Especially, it is worth noting that AutoNavi waterlogging map is no longer updated. There is a pressing demand to propose new monitoring system for urban waterlogging.

## 4.4 Nationwide map of urban waterlogging

We built a nationwide waterlogging map with more than 6,000 flood deposits based on microblogging data from 2017 to 2018, as shown in Figure 8. The locations of these flood deposits were manually located. The map was generated by ArcGIS 10.6 software. The small black dots represent the flood deposits (as shown in Figure 8a), and orange dots represent the number of the flood deposits within the province (as shown in Figure 8b). For example, the more flood deposits in the province, the larger the orange points. We notice an overall correlation between the economic development and the number of urban flood deposits. We used the GDP of each province as its background color in the map, i.e., the higher the GDP, the darker the color. It can be seen that the provinces located in eastern and central regions are more developed than northeast and western regions in China, as shown in Figure 8b. Also, these regions have higher population and Internet penetration rates than other regions. Accordingly, we found that the number of flood deposits in the eastern and central provinces is larger than northeast and western provinces. This phenomenon may be caused by the rapid urbanization and ground hardening so that the water permeability is greatly reduced. The potential economic loss in developed area is larger, and the real-time monitoring system for urban waterlogging disasters is more important in developed areas. On the other hand, the new technology adoption is also a factor related to the size of the orange dot. For example, Nanjing, Jiangsu Province has more than 200 posts every day, while Lhasa, Tibet Autonomous Region has about 40 posts per day. When a flood occurs, people discuss more waterlogging-related content on Weibo, we will collect more posts related to the waterlogging, which will lead to the orange dot bigger. Therefore, the residents of Nanjing area that discussed waterlogging via Weibo may make the orange dot bigger. However, it is worth noting that the new technology adoption itself is correlated to GDP.

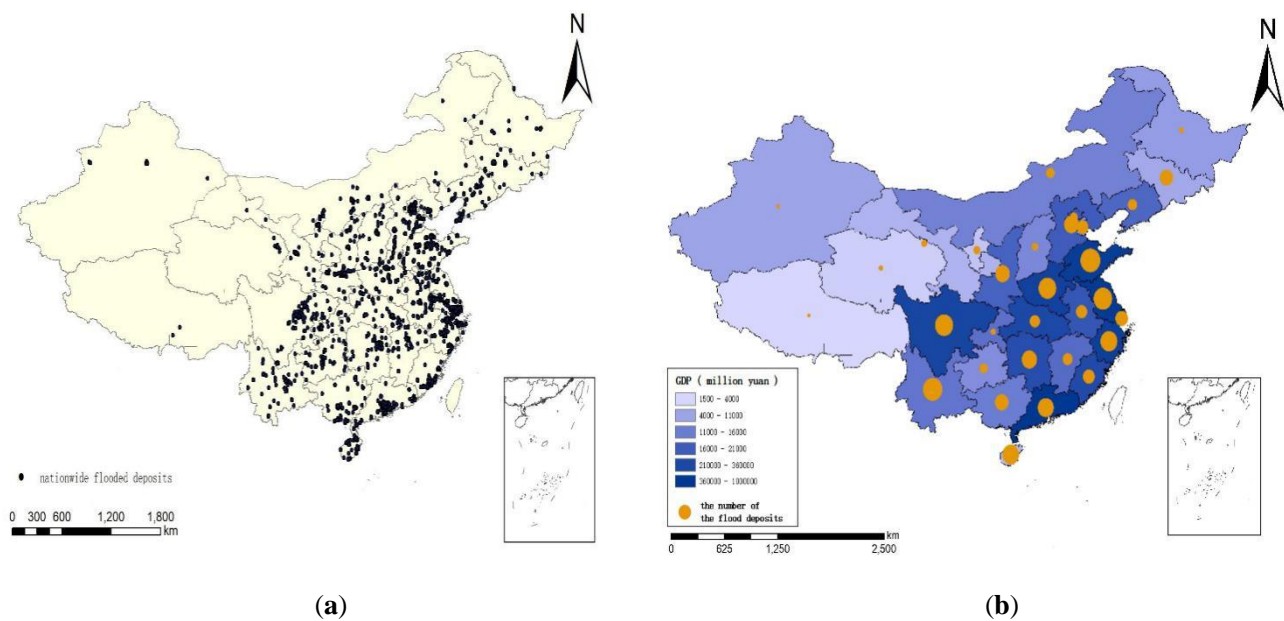

(**a**)                (**b**)

**Figure 8**: Nationwide map of waterlogging: flood deposits image (**a**), the number of the flood deposits image (**b**).

## 4.5 Monitoring system based on WeChat applet

To facilitate the usage of the nationwide map of urban waterlogging, we developed an applet based on WeChat, a popular mobile social app in China. The applet visualizes all urban waterlogging points in the map of China, as shown in Figure 9. Click on a waterlogging icon, a popup dialog will display the detail information of this urban waterlogging, such as the location description, longitude and latitude. Especially, the applet runs a daemon monitoring process that computes the distance of current position to waterlogging points nearby. In rainstorm weather, the applet would greatly benefit taxis and

bus drivers. When a driver approaches a flood deposit, the applet will make a voice alarm, such as "warning, warning, waterlogging ahead", to remind the driver that there is a flood deposit ahead and driving carefully.

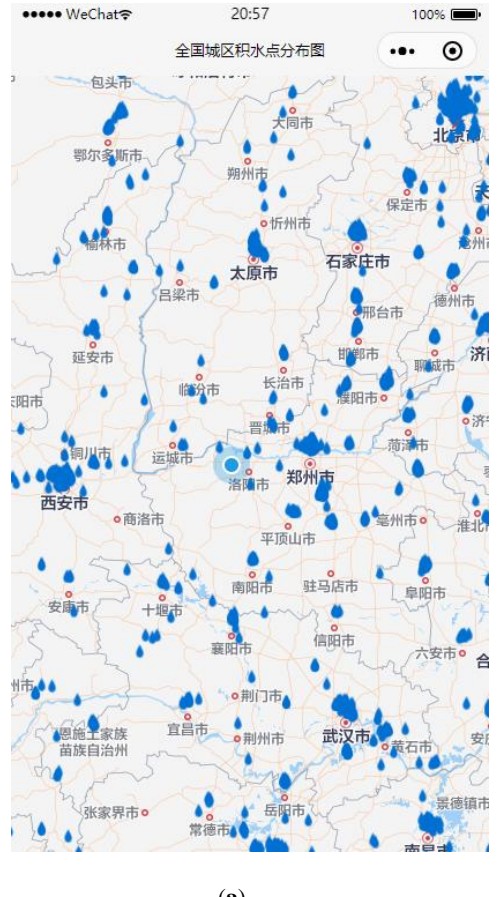 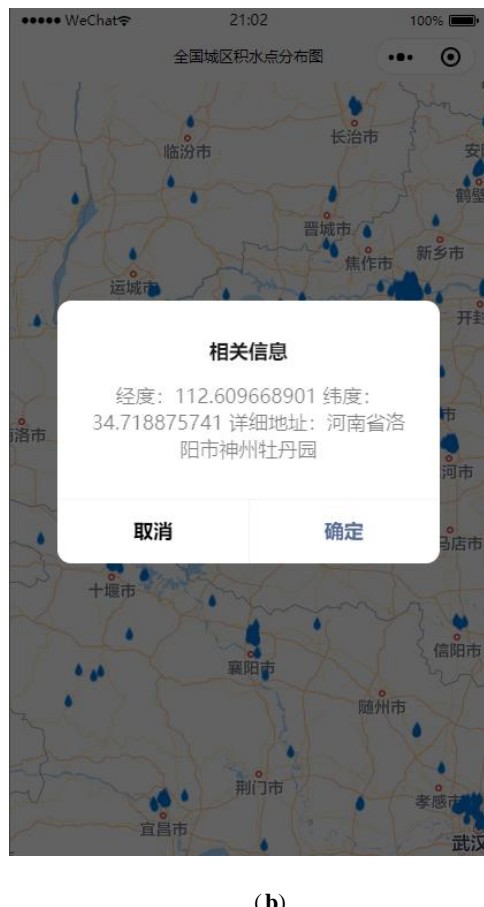

(**a**)                                                                                          (**b**)

**Figure 9:** Screenshots of monitoring system based on WeChat applet. (**a**) Nationwide map of urban waterlogging (© Tencent Map), and (**b**) detailed information of one waterlogging point.

## 5 Discussion

In this paper, we trained text classification models to identify microblogging text related to urban waterlogging caused by flood disasters. By manually collecting more than 4,400 waterlogging-related Weibo posts from 2017 to 2018, we built a gold-standard dataset to evaluate three text classification models, including RNN, SVM and GBDT. Our empirical experimental results showed that RNN model achieves higher accuracy than other two classifiers on an independent test set. Also, we found that the feature representation extracted by word2vec could improve performance compared to traditional TF-IDF feature. Furthermore, we built a nationwide map of urban waterlogging based on recent two-year microblogging data, and a monitoring system based on WeChat applet that will alert the user via voice alarm when he/she approaches a waterlogging point.

We have completed the development of the urban waterlogging monitoring system based on the WeChat applet, a very popular social media software similar to WhatsApp in Europe and American. We would like to release the applet as a plugin of WeChat, so that user can launch this application from WeChat by one click. With the help of WeChat's powerful web service capability and wide application, it is helpful for people to monitor the flood deposits, especially for taxis and bus drivers.

The limitation of our study lies in that the number of positive samples in the data set is relatively small. In future study, we set about to extend the scale of data set to build predictive model with better performance. Meanwhile, we will consider other deep learning models, such as convolutional neural networks (CNN), to further integrate remote sensing images and social media to improve the prediction of urban flood deposits and develop more powerful monitoring system of urban waterlogging.


**Author Contributions:** Hui Liu proposed the original idea; Hui Liu and Ya Hao designed and performed the experiments; Wenhao Zhang and Hanyue Zhang help to process data sets; Ya Hao draft the manuscript and Hui Liu revised the manuscript; Jinping Tong supervised this work and provided funding for this study. All authors have read and agreed to publish the manuscript.

**Funding:** This research was funded by National Natural Science Foundation of China (Grant number 91846203); Graduate Student
Scientific Research Innovation Projects in Jiangsu Province (Grant number KYCX20_2619, and KYCX20_2620).

**Data Availability:** The source code and manual collected Weibo posts that report flood deposits can be download from Github: https://github.com/hliu2016/waterlogging

**Conflicts of Interest:** The authors declare no conflict of interest.

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
