# Peer review of "Online Urban Waterlogging Monitoring Based on Recurrent Neural Network for Classification of Microblogging Text"

_Natural Hazards and Earth System Sciences, 2020_

## Referee Comment (RC1) · Anonymous Referee #1 · 2 Dec 2020

The article is well-structured although it might be good to do another language edit. I was interested to see that RNN networks work so well for Chinese text. While the methods applied in the article are not very new, this is indeed – to the best of my knowledge – the first dataset related to waterlogging specifically. However, there are a number of things that the authors should address.

L 19. Would be nice to go in a bit more detail here. Do the authors mean to say predict or detect? Online media is especially good at detecting events, but are very limited for predicting events.

L. 94. Is this location attached to every post? The localization for the waterlogging

seems to be done based on the text itself. Could the authors use the coordinates attached to the post, to deal with duplicate location names at a later stage. This could make the application described at the end of the paper more useful as less or no manual work is required depending on your response to my comment regarding L. 125. See for example: de Bruijn, J.A., de Moel, H., Jongman, B. et al. TAGGS: Grouping Tweets to Improve Global Geoparsing for Disaster Response. J geovis spat anal 2, 2 (2018). https://doi.org/10.1007/s41651-017-0010-6

L. 97. It would be nice for the non-Chinese reader to include some information on communities. For example, how many communities are there on average per city? If only 307 cities are included, could posts also be referring to cities that are not included in the city database and therefore not detected as duplicate?

L. 105. Are there more ambiguous cases, such as the threat of waterlogging, severe rainfall but no waterlogging etc.? Was there any ambiguity with real floods? If so, the authors should in more detail discuss how these posts tagged as relevant or not.

L. 125. If I correctly understand the authors 1) create a dataset with keywords "æůź" and "çğŕæřť", 2) manually select only those that are relevant to waterlogging and 3) select those that also mention a community. The positive samples are those that both mention a community and waterlogging, while the negative posts are only those that are not related to waterlogging. This means that the tweets that are related to waterlogging but do not mention a location are neither in the positive nor in the positive set.

Therefore, 1) A lot of manual work is required should the classifiers be applied to another dataset. This makes the automated classifier hard to apply "in the wild". 2) The classifier could in fact be learning location names rather than other features because location names are artificially inflated in the positive class.

L. 139. How were stop words determined?

[Figure]

L. 216. Recurrent neural networks and LSTMs are a relatively older neural network for text classification. Did the authors consider algorithms such as BERT and XML-RoBERTa. Usually these algorithms perform much better.

L. 402. The waterlogging posts seem to be manually located – or at least take manual work before deduplication – this should be mentioned in the discussion.

L. 410. The authors mention that a higher GDP indicates more waterlogging. However, provinces in the east also have a much higher population, and internet penetration. These factors need to be considered when making these claims.

L. 415. The labels in Figure 8 are Chinese.

L. 417. Could the authors validate the results, for example by comparing the results to significant rainfall events?

L. 428. Could the authors elaborate what needs to be done to apply the methods in this study in real-time (as the application shown in 4.5 only makes sense when applied in real-time)?

---

## Referee Comment (RC2) · Valerio Lorini (Referee) · 21 Dec 2020

The paper describes how information extracted from micro-blogging platform Weibo can be used to build a model for detecting urban floods. The model was trained using ground truth data merged with social media posts as training dataset and a set of known events have been used as reference for evaluation of the transfer learning potential of the model.

I think the paper is overall clear and well structured. While the work does not present a novelty in terms of technology or methodology, the effort of (i) applying it to a new, according to my knowledge, micro-blog data source with an extensive authoritative

ground truth and (ii) to a specific type of flood, namely urban waterlogging makes the paper interesting for community researchers. The literature review covers widely the related works but is missing some of the most recent developments where similar methodology was applied.

The methodology applied is correct and the results well clarified and documented with clear figures and tables. The authors made the code and data available, which is a much appreciated and very good practice.

General comments - Since the publisher is the European Geoscience Union (EGU) and the audience is expected to have little experience with Weibo, it would be useful to give some more context to non-users of the platform about how the original posts are structured and how the data was extracted in terms of text and location.

- While it is very clear and well explained how the model was trained and the data was prepared, little description of the operational Monitoring of urban waterlogging in real time was provided. The authors could elaborate more on the scalability of their system.

Minor comments L120 It is not clear to me why multiple posts actually located to the same flood deposit were removed. It means posts had same text and locations or different text and same location? in case could you explain better maybe with examples or by stating the relation between positive samples and location. Is it 1 to 1? or many to 1? or many to many? L120 It would be of great help to have the table 1 partially translated and described further, as it has been done in the following paragraph about the selection of negative samples. L210 How the undersampling was achieved? randomly removing posts or is there a methodology applied? L335 At some point the authors introduce the term 'flood deposit' and it seems to be used as an interchangeable term with waterlogging. Since the term is repeated extensively It would be clearer for reader to get the definition in the introduction of the paper when waterlogging is introduced. L300 Why not using an example referring to a flood post? L405 If numbers of events is based on microblogging, could it be that 'new technology adoption' rather than GDP

is a leading factor for bigger orange spots? i.e. there are more waterlogging because the population discuss more about it on weibo? in order to clear such doubt it could be useful to report the number of posts per day in the several areas.

Thanks

---

## Author Comment (AC1) · 29 Dec 2020

Dear Referee,

Thank you very much for giving us the opportunity to improve our manuscript. Upon the insightful and constructive comments given by you, we have conducted a careful and thorough revision of our manuscript. Before the presentation of our point-by-point responses to the comments given by you, we summarize the major revisions of the manuscript as follows:

1. We have introduced the effects of population and Internet penetration and revised

the manuscript accordingly.

2. We have translated the Chinese in Figure 8 to English.

3. We added an example of flooding to validate the results in WeChat applet.

4. We have revised our manuscript by following your suggestions.

These revisions, following the suggestions of you, have significantly improved the quality of our manuscript, and made our method more clearly to users. Once again, we sincerely thank you for the constructive comments.

Sincerely yours,

Hui Liu, Ya Hao, Wenhao Zhang, Hanyue Zhang, Fei Gao, Jinping Tong*

L. 19. Would be nice to go in a bit more detail here. Do the authors mean to say predict or detect? Online media is especially good at detecting events, but are very limited for predicting events.

Response: Thank you for your insightful comment. We can not agree with you any more that the social media is good at detecting events, but are very limited for predicting events. However, here we mean the prediction in general sense, namely, prediction of next disaster event by training an effective prediction model based on a collection of data sets. Take flood as example, we expect that the model can integrate various data resources, including weather forecast, historical waterlogging events and hydrogeological data, to predict the occurrence of flood and timely remind the residents to prepare for the disaster, thereby reduce the loss and achieve the early warning function.

L. 94. Is this location attached to every post? The localization for the waterlogging seems to be done based on the text itself. Could the authors use the coordinates attached to the post, to deal with duplicate location names at a later stage. This could make the application described at the end of the paper more useful as less or no manual work is required depending on your response to my comment regarding L. 125.

See for example: de Bruijn, J.A., de Moel, H., Jongman, B. et al. TAGGS: Grouping Tweets to Improve Global Geoparsing for Disaster Response. J geovis spat anal 2, 2 (2018). https://doi.org/10.1007/s41651-017-0010-6

Response: Yes, we agree with you very much that it is very convenient to use the coordinates attached to the posts to handle duplicate locations. However, not every post has attached coordinates, as mentioned in the example you gave, because the user's GPS location is turned off by default (de Bruijn et al., 2018). In fact, we found that there are only a few posts with attached coordinates. On the other hand, some posts related to urban waterlogging actually specified the locations of flooded deposits in the text, instead of the coordinates attached to the posts. For example, the post "Xingqing district, Yinchuan city,Shanghai road (Jinning street to FengHuang street) section because of the heavy rain, the road surface water was more serious." specified the waterlogging occurred in Shanghai road in the text of this post. In addition, Weibo users often "@" the official microblog of the local government department or the traffic police to remind, leading to mismatch between the true locations of waterlogging and the coordinates attached to the posts. In fact, we can identify the locations of the flood deposits by matching the catalogue of national-wide community. Therefore, the localizations of the waterlogging were done based on the textual content itself rather than the coordinates attached to the post.

L. 97. It would be nice for the non-Chinese reader to include some information on communities. For example, how many communities are there on average per city? If only 307 cities are included, could posts also be referring to cities that are not included in the city database and therefore not detected as duplicate?

Response: Thank you for your important suggestion. We have collected community information in 307 cities from Anjuke website, a popular housing website. Each city includes an average of 1,500 social communities. For the localization, we noticed that there are two cases to be deal with: one is that a community name contained in the microblog overlapped with other communities, namely, two or more communities

have the same name but located in different cities. The other is that the communities mentioned were not included in the database. For both cases, we exert to manual matching and deduplication. We manually checked the communities with duplicate names in different cities to ensure that the flood deposits were accurately located. Also, if a post referred to certain city not included in the catalog, we also manually identify the waterlogging locations by exploiting the community names or the coordinates attached to the post.

L. 105. Are there more ambiguous cases, such as the threat of waterlogging, severe rainfall but no waterlogging etc.? Was there any ambiguity with real floods? If so, the authors should in more detail discuss how these posts tagged as relevant or not.

Response: Thank you for your constructive suggestion. In the collection of Weibo posts associated to waterlogging, we used the Weibo API to obtain posts including the keywords "waterlogging". We suppose most of such posts state the fact that waterlogging events occurred in some places. However, in real life, more complex situations do occur, such as some places with heavy rainfall but no flooding, indicating that local drainage infrastructure and dewatering facilities works well and the residents were not threatened by flood. On the other hand, there were many disease-related posts that are not related to urban waterlogging as we discussed in the manuscript. To exclude these specious posts, we manually marked relevant and irrelevant based on whether the content of the posts were related to urban waterlogging.

L. 125. If I correctly understand the authors 1) create a dataset with keywords "æĚŽuʹz" and "çgĚŸʹ ræĚĞ rt'", 2) manually select only those that are relevant to waterlogging and 3) select those that also mention a community. The positive samples are those that both mention a community and waterlogging, while the negative posts are only those that are not related to waterlogging. This means that the tweets that are related to waterlogging but do not mention a location are neither in the positive nor in the positive set. Therefore, 1) A lot of manual work is required should the classifiers be applied to another dataset. This makes the automated classifier hard to apply "in the wild". 2) The

classifier could in fact be learning location names rather than other features because location names are artificially inflated in the positive class.

Response: Thank you for your comment. In fact, we created a dataset with keywords "drowning" and "waterlogging" so that the posts unrelated to waterlogging are excluded. Next, we selected posts that mention not only communities, but also other locations, such as roads, schools, etc. So, the positive samples are exactly those posts that include both locations and waterlogging. However, the negative samples consist of all remaining posts, including those irrelevant to waterlogging, as well as the posts with waterlogging keywords but without specific locations in the text. In fact, we aim to build a nationwide flood map and a monitoring system based on WeChat applet, posts related to waterlogging but not including locations were excluded because the exact location could not be specified. In addition, it is worth noting that a huge number of posts irrelevant to waterlogging actually include a lot of location names, and these posts are used as negative samples to train the classifier. So, we believe that the classifier actually learn the waterlogging–related features of posts, and the location names are not artificially inflated.

L. 139. How were stop words determined?

ResponseïijŽThank you for your comment. In the word-based retrieval system, words with high frequency but without retrieval significance are determined as stop words (Guan et al., 2017), such as "of", "is", etc. We complete this task by exploiting the stop word list released by Harbin Institute of Technology (Guan et al., 2017), which is a widely used stop word catalogue. Also, we removed the words that appeared frequently in posts but had nothing to do with the classification task, for example "Weibo", "video", etc.

L. 216. Recurrent neural networks and LSTMs are a relatively older neural network for text classification. Did the authors consider algorithms such as BERT and XML-RoBERTa. Usually these algorithms perform much better.

Response: Thank you for your constructive suggestion. BERT, which stands for Bidirectional Encoder Representations from Transformers, pre-trained deep bidirectional representations by jointly conditioning on both left and right context in all layers (Wu et al., 2019). In the selection the classification algorithms, we actually considered BERT, as well as XMLRoBERTas. However, due to the imbalance of positive and negative samples in the training dataset, the number of negative samples intensively surpasses that of positive samples. When using BERT algorithm, we did not find an appropriate way to sample the negative samples set for multiple times to ensure that each negative sample could be seen by the classifier. Although the LSTM model achieves superior performance, as illustrated in our evaluation experiments, we really appreciate your constructive suggestion and plan to adopt the BERT model in our future work.

L. 402. The waterlogging posts seem to be manually located – or at least take manual work before deduplication – this should be mentioned in the discussion.

Response: Thank you for your important suggestion. We have modified the manuscript to show that the waterlogging posts were manually located.

L. 410. The authors mention that a higher GDP indicates more waterlogging. However, provinces in the east also have a much higher population, and internet penetration. These factors need to be considered when making these claims.

Response: Thank you for your constructive suggestion. We have described these factors of population and internet penetration in the eastern provinces, and modified the manuscript accordingly.

L. 415. The labels in Figure 8 are Chinese.

Response: Thank you for your careful review of our manuscript. We have modified the labels in Figure 8.

L. 417. Could the authors validate the results, for example by comparing the results to significant rainfall events?

Response: Thank you for your suggestion. In rainstorm weather, the applet would benefit taxi and bus drivers. Because when a driver approaches a flooded deposit, the applet will make a voice broadcast, reminding the driver that there is a flooded deposit ahead and driving carefully. For example, in July 2020, Kunming, Yunnan province, was hit by a rainstorm, resulting in serious water at the crossroad between the Haiyuan Middle Road and Keyi Road in Wuhua District. Our applet will automatically calculate the distance between the current location and this crossroad, provided the GPS is turned on. When the distance to the crossroad is less than a predefined threshold, our applet trigger voice alarm, such as "warning, warning, waterlogging ahead", to remind drivers.

L. 428. Could the authors elaborate what needs to be done to apply the methods in this study in real-time (as the application shown in 4.5 only makes sense when applied in real-time)?

Response: Thank you for your import suggestion. We have completed the development of the urban waterlogging monitoring system based on the WeChat applet, a very popular social media software similar to WhatsApp in Europe and American. We would like to release the applet as a plugin of WeChat, so that user can launch this application from WeChat by one click. With the help of WeChat's powerful web service capability and wide application, it is helpful for people to monitor the flood deposits, especially for taxis and bus drivers.

Reference

de Bruijn, J.A., de Moel, H., Jongman, B. et al.: TAGGS: Grouping Tweets to Improve Global Geoparsing for Disaster Response, J geovis spat anal, 2, 2, doi: https://doi.org/10.1007/s41651-017-0010-6, 2018. Guan, Q., Deng, S., Wang, H.: Chinese Stopwords for Text Clustering: A Comparative Study, Data Analysis and Knowledge Discovery, 1(3), 72-80, 2017. Wu, X., Lv, S., Zang, L., et al.: Conditional BERT contextual augmentation, International Conference on Computational Science,

[Figure]

Springer, Cham, 84-95, 2019.

Please also note the supplement to this comment:
https://nhess.copernicus.org/preprints/nhess-2020-335/nhess-2020-335-AC1-supplement.pdf

---

## Author Comment (AC2) · 29 Dec 2020

Dear Valerio Lorini,

Thank you very much for giving us the opportunity to improve our manuscript. Upon the insightful and constructive comments given by you, we have conducted a careful and thorough revision of our manuscript. Before the presentation of our point-by-point responses to the comments given by you, we summarize the major revisions of the manuscript as follows: 1. We have added the most recent progress in the text classification using the recurrent neural network. 2. We have revised our manuscript according to your comments and suggestions.

We believe these revisions have significantly improved the quality of our manuscript, and made our method more clearly to users. Once again, we sincerely thank you for the constructive comments.

Sincerely yours,

Hui Liu, Ya Hao, Wenhao Zhang, Hanyue Zhang, Fei Gao, Jinping Tong*

The paper describes how information extracted from micro-blogging platform Weibo can be used to build a model for detecting urban floods. The model was trained using ground truth data merged with social media posts as training dataset and a set of known events have been used as reference for evaluation of the transfer learning potential of the model.

I think the paper is overall clear and well structured. While the work does not present a novelty in terms of technology or methodology, the effort of (i) applying it to a new, according to my knowledge, micro-blog data source with an extensive authoritative ground truth and (ii) to a specific type of flood, namely urban waterlogging makes the paper interesting for community researchers. The literature review covers widely the related works but is missing some of the most recent developments where similar methodology was applied.

Response: Thank you for your constructive suggestion. In our manuscript, we have added the most recent progress in text classification using recurrent neural network.

The methodology applied is correct and the results well clarified and documented with clear figures and tables. The authors made the code and data available, which is a much appreciated and very good practice.

General comments - Since the publisher is the European Geoscience Union (EGU) and the audience is expected to have little experience with Weibo, it would be useful to give some more context to non-users of the platform about how the original posts are structured and how the data was extracted in terms of text and location.

Response: Thank you for your comment. Weibo is run by sina.com Company in China and it is similar to Twitter. A Weibo post often contains the fields such as user ID, user name, microblog content, posted time, etc. Some posts include pictures. We used keywords "drowning" or "waterlogging" to download posts using Sina Weibo API, as such posts often describe waterlogging events already occurred in some places. To identify the location of waterlogging, we first marked the names of communities, roads and orientation from the posts using named entity recognition technique. Next, the names were matched to the national geographic names and national community names to extract the accurate geographic location.

- While it is very clear and well explained how the model was trained and the data was prepared, little description of the operational Monitoring of urban waterlogging in real time was provided. The authors could elaborate more on the scalability of their system.

Response: Thank you for your suggestion. We have completed the development of the urban waterlogging monitoring system based on the WeChat, which is a social media software similar to WhatsApp in Europe and American. Our applet can be launched by one click from WeChat. Once the applet is started, it will automatically calculate the distance from the current location to the nearby waterlogging point. When the distance exceeds the predefined threshold, the applet will trigger voice alarm to remind drivers that there is a waterlogging point nearby and drive carefully. With the help of WeChat's powerful web service capability and wide application, it is helpful for a lot of people to monitor flood deposits, especially for taxi and bus drivers.

Minor comments:

L120 It is not clear to me why multiple posts actually located to the same flood deposit were removed. It means posts had same text and locations or different text and same location? in case could you explain better maybe with examples or by stating the relation between positive samples and location. Is it 1 to 1? or many to 1? or many to many?

Response: Thanks for your insightful comments. We deleted duplicate posts that have same text and same location. If their content is different, we will keep all, even if their location is same. For example: (i) On the morning of May 12th, the reporter learned from the Nanning Traffic Police Detachment that because of the rainy day leading to Pingle Avenue surrounding roads have part of water. (ii) At present, there have been water accumulation in some sections of Yinhai Avenue, Ruihejiayuan and Pingle Avenue, please pedestrians and vehicles are requested to pass carefully. We kept both posts. If the content of these two posts was similar or the same, then we would keep one of them.

L120 It would be of great help to have the table 1 partially translated and described further, as it has been done in the following paragraph about the selection of negative samples.

Response: Thank you for your important suggestion. We have translated Table 1 in the revised manuscript.

L210 How the undersampling was achieved? Randomly removing posts or is there a methodology applied?

Response: The undersampling is done by using the third-party library imblearn implemented in Python. The RandomUnderSampler function in imblearn package implements boostrap sampling by setting the parameter replacement to true. The function randomly remove samples in original dataset with multiple classes, to build the balanced subset of the whole samples.

L335 At some point the authors introduce the term 'flood deposit' and it seems to be used as an interchangeable term with waterlogging. Since the term is repeated extensively it would be clearer for reader to get the definition in the introduction of the paper when waterlogging is introduced.

Response: Thank you for your careful review of our manuscript. "Flood deposit" and

"waterlogging" are interchangeable in most cases. However, "Flood deposit" is more sense of a geographical location, which is easy to be flooded in the case of rainstorm. "Waterlogging" is more sense that flooding events occurred.

L300 Why not using an example referring to a flood post?

Response: Thank you for your constructive suggestion. In the manuscript, we have modified this part as follows: For example, a post reads as follows: In the early hours of this morning, there was a heavy rain in Shenzhen. Many roads were flooded, and only the top of cars could be seen. Among them, the water at the bottom of Hezhou Bridge was serious, which caused the road to be interrupted. TP means true positive: the true is positive, and the classifier predicts a positive. For example, Bottom of Hezhou Bridge is flooded, and the classifier accurately reports this. TN means true negative: the truth is negative, and the classifier predicts a negative. For instance, Bottom of Hezhou Bridge is not flooded, and the classifier accurately reports this. FP means false positive: the truth is negative, but the classifier predicts a positive. Such as Bottom of Hezhou Bridge is not flooded, but the classifier inaccurately reports that it is. FN means false negative: the truth is positive, but the classifier predicts a negative. For example, Bottom of Hezhou Bridge is flooded, but the classifier inaccurately reports that is not.

L405 If numbers of events is based on microblogging, could it be that 'new technology adoption' rather than GDP is a leading factor for bigger orange spots? i.e. there are more waterlogging because the population discuss more about it on weibo? in order to clear such doubt it could be useful to report the number of posts per day in the several areas.

Response: Thank you for your insightful comment. We agree with you very much that new technology adoption is a factor related to the size of the orange dot. For example, Nanjing, Jiangsu Province has more than 200 posts every day, while Lhasa, Tibet Autonomous Region has about 40 posts per day. When a flood occurs, people discuss more waterlogging-related content on Weibo, we will collect more posts related to the

waterlogging, which will lead to the orange dot bigger. Therefore, the residents of Nanjing area that discussed waterlogging via Weibo may make the orange dot bigger. However, it is worth noting that the new technology adoption itself is correlated to GDP. We will carry out more analysis to explore the factors related to the number of Weibo posts discussing waterlogging in the future work.

Please also note the supplement to this comment:
https://nhess.copernicus.org/preprints/nhess-2020-335/nhess-2020-335-AC2-supplement.pdf

---

## Author Response (AR2)

Dear Editor:

Thank you very much for giving us the opportunity to improve our manuscript. Upon the insightful and constructive comments given by the reviewer, we have made revision to our manuscript. We have presented our point-to-point responses to the comments given by the reviewer.

L55

>>Their RNN model adopts word embedding techniques, including TF-IDF and n-gram, to capture broad context of the words in social media text for better classification (Amin et al., 2020). Jelodar et al. used automated extraction of the novel coronavirus-related posts from social media and topic modeling to uncover various issues related to COVID-19 from public opinions, and then investigated how to use LSTM recurrent neural 60network for sentiment classification of COVID-19 comments (Jelodar et al., 2020). <<

I appreciate the effort of adding some reference to recent developments in Text Classification but please, there is no need to pick references related to coronavirus. There is plenty of reference available for Text Classification within Disaster Risk Management.

Response: Thank you for your important suggestion. We revise this part as below: Wadawadagi et al. investigated the severity of disaster events from micro-blog messages during natural calamities and emergencies using convolutional neural networks (CNN) and recurrent neural networks (RNN). Their work employed a joint model to combine the features of CNN with RNN, taking account of the coarse-grained local features generated via CNN and long-range dependencies learned through RNN for analysis of small text messages (Wadawadagi et al., 2020). Also, Singh et al. investigated the problem of localization using the social sensing model (Twitter) to provide an efficient, reliable and accurate flood text classification model with minimal labeled data. They proposed to perform text classification using the inductive transfer learning method for effective classification of flood-related feeds in new locations (Singh et al., 2020).

L475

>>We have completed the development of the urban waterlogging monitoring system based on the WeChat applet, a very 475popular social media software similar to WhatsApp in Europe and American. We would like to release the applet as a plugin of WeChat, so that user can launch this application from WeChat by one click. With the help of WeChat's powerful web service capability and wide application, it is helpful for people to monitor the flood deposits, especially for taxis and bus drivers.<<

I'd suggest to remove the above text from the conclusions as the app is not a contribution to research (and if it is it should be explained how). Authors already wrote about the app in a specific paragraph.

Response: Thank you for your constructive suggestion. We have removed the above text from the conclusions.

These revisions, following the suggestions of the reviewer, have significantly improved the quality of our manuscript, and made our method more clearly to users. Once again, we sincerely thank the reviewers for the constructive comments, and thank you for considering our paper as a candidate for publication in Natural Hazards and Earth System Sciences.

Sincerely yours,

Hui Liu, Ya Hao, Wenhao Zhang, Hanyue Zhang, Fei Gao, Jinping Tong*